# GradSim: Gradient-Based Language Grouping
# for Effective Multilingual Training

**Mingyang Wang**[1,2]  **Heike Adel**[3]  **Lukas Lange**[1]
**Jannik Strötgen**[4]  **Hinrich Schütze**[2]
[1]Bosch Center for Artificial Intelligence, Renningen, Germany
[2]LMU Munich, Germany  [3]Hochschule der Medien, Stuttgart, Germany
[4]Karlsruhe University of Applied Sciences, Germany
mingyang.wang2@de.bosch.com

## Abstract

Most languages of the world pose low-resource challenges to natural language processing models. With multilingual training, knowledge can be shared among languages. However, not all languages positively influence each other and it is an open research question how to select the most suitable set of languages for multilingual training and avoid negative interference among languages whose characteristics or data distributions are not compatible. In this paper, we propose GradSim, a language grouping method based on gradient similarity. Our experiments on three diverse multilingual benchmark datasets show that it leads to the largest performance gains compared to other similarity measures and it is better correlated with cross-lingual model performance. As a result, we set the new state of the art on AfriSenti, a benchmark dataset for sentiment analysis on low-resource African languages. In our extensive analysis, we further reveal that besides linguistic features, the topics of the datasets play an important role for language grouping and that lower layers of transformer models encode language-specific features while higher layers capture task-specific information.

## 1 Introduction

Most natural language processing (NLP) research today still focuses on a small number of languages. Extending NLP models to further languages poses different challenges, i.a., little (annotated) data (Hedderich et al., 2021). Multilingual training can help in those cases by sharing knowledge across languages. However, adding new languages to the multilingual training set may not necessarily lead to performance gains. In fact, certain languages might actually hurt the performance on downstream tasks in a specific target language (Adelani et al., 2022; Snæbjarnarson et al., 2023), for instance, due to unrelatedness to the target language.

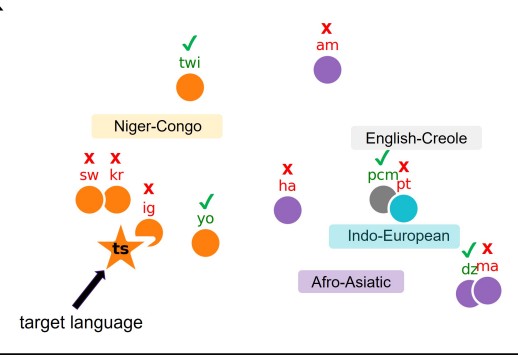

Figure 1: Exemplary transfer learning setup with African languages: The motivation for this work is that neither language family (indicated by node colors) nor typological distance (indicated by distance of languages in the plot) are consistent predictors of good performance when choosing a source language for cross-lingual transfer. Red cross: source language affecting performance negatively. Green tick: source language affecting performance positively.

As a solution, previous work investigates different measures for language similarity and selects only languages similar to the target language for the multilingual training set (i.a. Tan et al., 2019; Lin et al., 2019; Pires et al., 2019; Oncevay et al., 2020; Shaffer, 2021; Snæbjarnarson et al., 2023). However, it is an open research question whether language similarity translates into performance gains of multilingual models. For multilingual training, other characteristics might play a role, such as topical shifts of the training data. As a result, it is still unclear how to select the set of languages that leads to the most effective multilingual training setup.

In this paper, we study multilingual fine-tuning of language models with a diverse set of training languages.[1] In particular, we show that linguistics-based language similarities are only weakly correlated with cross-lingual transfer performance. Figure 1 illustrates a sample case in which neither

---

[1]Note that our proposed method is generic and could also be applied for multilingual pre-training.

language family information (indicated by node colors) nor similarity of language embeddings (indicated by proximity in the vector space) is helpful for finding languages that have a positive cross-lingual transfer score with the target language. Thus, prior information about the languages, such as their language families or typological features, alone is not enough for an effective multilingual training. Instead, similarity measures that capture additional information about the data and task beyond linguistics similarity may achieve better performance. Wang et al. (2020), for instance, show that gradient similarity across languages measured along the optimization trajectory correlates with language proximity and cross-lingual performance.

We draw inspiration from this observation. However, instead of projecting conflicting gradients throughout the training process, we propose to leverage the *gradient similarity* to group languages with a branch-and-bound-like algorithm that *optimizes the overall similarity score* of all languages. This approach has the following advantages: (i) It can be applied *without any prior knowledge* of the languages or topics of the given datasets, (ii) it is *well correlated* with downstream task performance of the multilingual model, (iii) it finds the best language groups from a *global perspective*, i.e., instead of selecting source languages independently of each other (which may create groups of mutually interfering languages), we form each group based on a criterion that evaluates the group as a whole.

In our experiments, we show the superior performance of our grouping method compared to various baseline approaches on three multilingual datasets with different tasks and set the new state of the art on AfriSenti, a sentiment analysis dataset in 12 low-resource African languages.

Furthermore, we extensively analyze our models with a topic analysis, a correlation-based analysis and an ablation study, revealing important insights, for instance that the topic distribution of the training data heavily affects multilingual training and that lower layers of transformer models encode language-specific features while higher layers capture task-specific information. This confirms results from prior work (i.a., Raganato and Tiedemann, 2018; Jawahar et al., 2019; Tenney et al., 2019; Kovaleva et al., 2019) from another (correlation-based) perspective.

The code base for `GradSim` is available online.[2]

## 2 Related Work

**Multilingual and multi-task training.** A growing number of research projects investigates multilingual training to cover a variety of languages, including low-resource languages (Hu et al., 2020; Lange et al., 2020; Hedderich et al., 2021; FitzGerald et al., 2022). In the context of low-resource sentiment analysis, Wang et al. (2023) recently use the cross-lingual transfer score between pairs of languages to select source languages for multilingual training. Our approach differs from these works in that we investigate language interactions from a global optimization perspective.

Considering each language as a separate task, multilingual training can be treated as a multi-task learning (MTL) problem (Ruder, 2017). A line of existing work utilizes gradient-based techniques to improve multi-task learning (Chen et al., 2018; Sener and Koltun, 2018; Yu et al., 2020; Wang et al., 2020). They show that negative cosine similarity between gradients leads to negative interference for MTL optimization, and projecting out the conflicting gradients can improve the optimization dynamics. Our work follows this insightful observation. However, in contrast to their work, we propose to leverage multilingual gradients for language grouping to ensure that gradients are aligned in each language group.

**Language similarity measures.** In order to group languages for multilingual training or transfer learning, related work has proposed different ways to estimate the similarity between languages, e.g., leveraging the language family taxonomy (Tan et al., 2019; Shaffer, 2021; Chronopoulou et al., 2023; Snæbjarnarson et al., 2023) or representing languages as information-rich vectors based on their typological or conceptual features (Littell et al., 2017; Lin et al., 2019; Oncevay et al., 2020; Liu et al., 2023).

Another line of works measures language similarity based on embeddings from multilingual pretrained language models (mPLMs) (Raganato and Tiedemann, 2018; Lange et al., 2021b; Chang et al., 2022; Lin et al., 2023). Tan et al. (2019) and Shaffer (2021), for instance, perform language grouping for multilingual named entity recognition and neural machine translation based on embeddings. In contrast to these studies, we propose to use the gradient cosine similarity between languages as the similarity measure for language grouping. This

model-based similarity measure reflects how each language interacts in the optimization process, with no need of any prior knowledge of the languages.

## 3 Method

In this section, we describe our proposed language grouping approach and the general multilingual training in which we apply its results. Note that the gradient-based similarity estimation is purely model-based, thus, can be applied to other settings, e.g., multi-domain or multi-task problems, as well.

### 3.1 Step I: Gradient Similarities

Due to the high discrepancy among languages, multilingual optimization often suffers from the conflicting gradient issue (Wang et al., 2020; Xu and Murray, 2022), i.e., gradients of different languages point into different directions. Previous works show that gradient similarity is correlated with model performance (Chen et al., 2018; Sener and Koltun, 2018; Wang et al., 2020; Yu et al., 2020). Inspired by this observation, we propose to use gradient similarity for grouping languages.

Given a set of languages $L = \{l_1, l_2, \ldots, l_N\}$, we study the gradient similarities across languages by training a multilingual model jointly on all languages and measure the language gradients $G = \{g_1, g_2, \ldots, g_N\}$ along the optimization process. To reduce computational costs, we average language gradients first at the epoch level and calculate the per-epoch gradient cosine similarity between languages. Then we average the gradient similarity over all epochs. Finally, we get a gradient similarity matrix $S \in \mathbb{R}^{N \times N}$ across $N$ languages, with $s_{i,j} = \cos(g_i, g_j) = \frac{g_i \cdot g_j}{|g_i||g_j|}$.

Since it is very expensive to calculate the gradient similarity based on the gradients w.r.t. all parameters, we choose to only use the gradients based on the classification layer of the model. An analysis of gradients of different layers and ablations studies can be found in Sections 5.2 and 5.3.

### 3.2 Step II: Language Grouping

Based on the pairwise similarity matrix $S$ from Step I, we next determine the best grouping into a pre-defined number of $K$ groups.

In particular, our goal is to find the $K$ language groups which (i) cover all languages of the given language set $L$, and (ii) maximize the overall similarity score of all languages, which is a reduction from the Set-Cover problem. We solve it using the

branch-and-bound-like algorithm as in Standley et al. (2020) and Fifty et al. (2021).[3] The algorithm evaluates different combinations of $K$ language groups under the constraint that each language is included in at least one, but potentially multiple groups. We finally select the language grouping that leads to the highest overall similarity score.

Given $\Gamma = \{\gamma_1, \ldots, \gamma_K\}$ as a potential grouping result, we define the overall similarity score for $\Gamma$ as $\sum_{i=1}^{N} \text{Sim}(l_i|\Gamma)$ where $\text{Sim}(l_i|\Gamma)$ is the collective similarity score of language $l_i$ in its language group $\gamma_j \in \Gamma$. The collective similarity score of $l_i \in \gamma_j$ is defined as the average of all pair-wise similarities between $l_i$ and the other languages in $\gamma_j$.

### 3.3 Steps III : Training and Inference

Given the language groups $\Gamma$ from Step II, we train one multilingual model per group $\gamma_j \in \Gamma$, using the training data from the respective languages. For inference, we select the appropriate multilingual model for each target language and apply it to the test data. If a target language $l_i$ appears in more than one group, we select the group with the highest collective similarity score of $l_i$ for inference.

## 4 Experiments

In this section, we describe our experimental settings as well as our results for three tasks.

### 4.1 Tasks and Datasets

We experiment with the following three datasets covering different languages as well as text classification and sequence tagging tasks. (Dataset statistics are given in Table 8 and 9 in Appendix A.)

**AfriSenti** (Muhammad et al., 2023a,b): This shared task dataset provides a challenging testbed for sentiment analysis: Both the languages (12 African languages) and the text genre (Twitter) pose challenges to NLP models. To investigate multilingual training results, we focus on the multilingual subtask of the shared task (Subtask B), and report macro-weighted F1 scores following Muhammad et al. (2023b).

**WikiAnn** (Pan et al., 2017): This dataset offers automatically extracted labels for named entity recognition (NER). Following Shaffer (2021), we select 15 languages for our experiments and use micro-averaged F1 as the evaluation metric.

---

[3]Alternatively, the binary integer program (BIP) solver could be used as in Zamir et al. (2018).

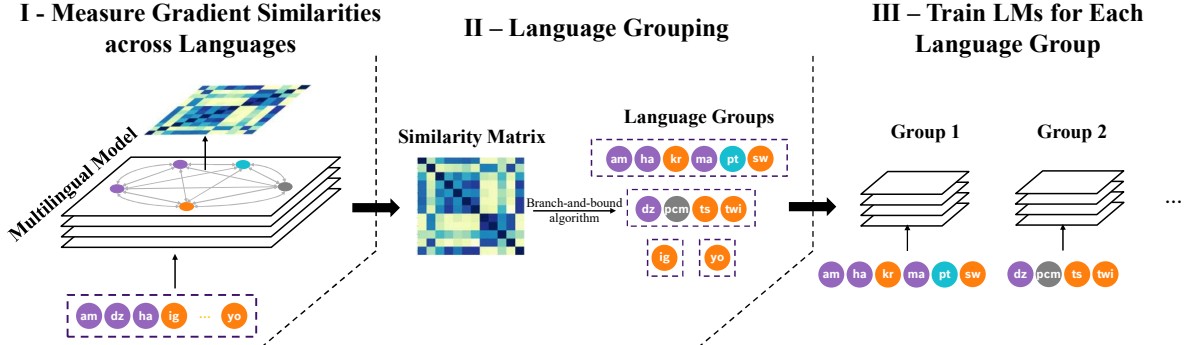

Figure 2: Overview of our proposed method GradSim for language grouping. (Step I) We train all languages in one multilingual model to measure the gradient similarities across languages. (Step II) We determine the best $K$ language groups based on the similarity measure from the first step. (Step III) We train one language model on each language group and deploy it for inference.

**Universal Dependency (UD) treebank v1.2** (Nivre et al., 2016): We experiment with part-of-speech (POS) tagging using the 17 Universal POS labels. Following prior work (Yasunaga et al., 2018; Lange et al., 2021a), we use 27 languages from the dataset with 21 high-resource and 6 low-resource languages and report accuracy for evaluation.

### 4.2 Training Details

For our experiments on AfriSenti, we use the pre-trained AfroXLM-R large transformer (Alabi et al., 2022), an XLM-R model adapted to African languages, as our base model. To measure language gradients in Step I, we use 25% of the training data in AfriSenti and set the batch size to 8 for computational efficiency. For multilingual training and inference (Step III), we use all training data and a batch size of 32. In both stages, we finetune the model with a learning rate of 1e-5 and set the maximum sequence length to 128. We set the number of language groups to $K = 4$ which equals the number of language families in the dataset. We further provide a comparison of different $K$ in Section 5.3.

For the NER task, we follow the training setup used in Shaffer (2021) for a fair comparison. Specifically, we use XLM-R as our base model and finetune it for 3 epochs. We set the batch size to 20 with a learning rate of 2e-5 and a max sequence length of 300. Following Shaffer (2021), we set the number of language groups to $K = 4$.

For the POS tagging task, we use the XLM-R model as well and set the training epoch to 20. We use a batch size of 8, a learning rate of 2e-5 and a maximum sequence length of 128. Here, we specify $K = 6$ for language grouping, as 6

language families are covered by the 27 languages we study.

On all three datasets, we use the AdamW optimizer (Loshchilov and Hutter, 2017). The training was performed on Nvidia A100 GPUs.[4] All reported results are averaged over 5 random seeds.

### 4.3 Baselines

Besides a **monolingual model** (trained only on the target language) and a purely **multilingual model** (trained on *all* available languages), we consider the following baselines for language grouping that have been presented by prior work:

**Language family.** We group languages based on their language family information and train one multilingual model per language family. Language family-based grouping is also studied by, i.a., Tan et al. (2019); Shaffer (2021); Chronopoulou et al. (2023); Snæbjarnarson et al. (2023).

**Typological similarity.** Languages can be represented by typological features, e.g., the syntax, phonology or inventory features. Using the *lang2vec* tool and the URIEL knowledge base (Littell et al., 2017), we retrieve language vectors and use the pairwise distances among them as the similarity measure of our algorithm. This is similar to Lin et al. (2019) and Oncevay et al. (2020).

**Embedding distance.** Multilingual pretrained language models (mPLMs) also encode language-specific information (Raganato and Tiedemann, 2018; Chang et al., 2022). Tan et al. (2019) and Shaffer (2021) use mPLM-based language embeddings to determine language similarities for language grouping. Following this idea, we compute

---

[4] All experiments ran on a carbon-neutral GPU cluster.

| Method | avg* | am* | dz | ha | ig* | kr* | ma* | pcm | pt* | sw* | ts | twi | yo* |
|---|---|---|---|---|---|---|---|---|---|---|---|---|---|
| Oracle upper bound | 71.46 | 69.87 | 69.45 | 80.15 | 78.48 | 70.72 | 52.41 | 68.67 | 71.85 | 61.59 | 55.42 | 64.79 | 75.52 |
| Multilingual | 59.97 | 51.99 | 56.62 | 66.10 | 67.64 | 61.03 | 43.07 | 55.70 | 67.37 | 58.18 | 42.86 | 49.94 | 62.87 |
| Monolingual | 68.29 | 49.00 | 57.16 | **80.36** | **79.55** | 70.72 | 48.73 | 68.24 | 66.12 | 62.50 | 44.66 | 54.56 | **75.09** |
| Language family | 66.19 | 67.84 | 68.54 | 78.88 | 68.16 | 61.10 | 51.10 | 67.32 | 64.88 | 58.38 | 47.04 | 54.60 | 64.36 |
| Typological similarity | 68.93 | 66.10 | **69.27** | 79.38 | 78.00 | 70.09 | 49.17 | 63.81 | 66.12 | **63.23** | 46.79 | 61.25 | 73.95 |
| Embedding dis. (PLM) | 68.81 | **72.09** | 67.20 | 71.57 | 74.49 | 71.89 | 51.49 | 69.09 | 67.32 | 62.50 | 52.01 | 60.88 | 75.09 |
| Embedding dis. (FT) | 69.62 | 59.69 | 67.36 | 79.78 | 78.71 | 69.99 | 50.01 | 68.46 | 66.43 | 62.62 | 49.20 | **64.01** | 75.07 |
| GradSim (ours) | **71.34** | 66.11 | 67.90 | 79.97 | 79.55 | 72.12 | **53.68** | 68.40 | **72.30** | 63.05 | 50.36 | 63.46 | 75.09 |

Table 1: Results on AfriSenti, a benchmark dataset for sentiment analysis on low-resource African languages. Bold shows best results, underline highlights second-best results per language and on average. * indicates the settings with statistically significant improvements ($p\text{-}value < 0.05$) using GradSim compared to Embedding dis. (FT), the overall second-best system.

| Method | Average F1 |
|---|---|
| SOTA Single model | 74.08 |
| GradSim+TAPT Single model (ours) | **75.29** |
| SOTA Ensemble | 75.06 |
| GradSim+TAPT Ensemble (ours) | **75.34** |

Table 2: Results on AfriSenti in comparison to the state of the art (Wang et al., 2023). We apply task-adaptive pretraining (TAPT) and ensemble methods on top of GradSim for a fair comparison to the state of the art.

sentence embeddings using the pretrained encoder from our base model and average sentence embeddings of the same language. Then, we use the embedding distance across languages as the similarity measure in Step I (denoted by Embedding distance (PLM)). As an alternative, we also consider embeddings from the language model fine-tuned on the task (denoted by Embedding distance (FT)).

**Oracle upper bound.** As an upper bound, we group languages based on the post-hoc cross-lingual transfer performance. The cross-lingual transfer performance is often used for source language selection as in Adelani et al. (2022) and Wang et al. (2023). We consider this an oracle upper bound as it is a direct indication of how knowledge learned from one language affects the performance on another language. Note that this approach is computationally expensive as it requires $N \times N$ transfer experiments for $N$ languages, while our gradient-based approach only needs a single training run for collecting gradient information.

## 4.4 Results

**Text classification.** Table 1 shows our experimental results on the AfriSenti dataset (per language and on average). While for a few languages, a grouping based on our baseline approaches performs best (e.g., embedding distance for *am* and

*pcm*, or typological similarity for *sw*), GradSim performs best or second best for most languages and, as a result, best on average. Its average result comes very close to the oracle upper bound, which, in contrast to our approach, requires prior knowledge about cross-lingual transfer performance.

We also compare GradSim with the state-of-the-art method on AfriSenti (Wang et al., 2023), which uses AfroXLM-R with task-adaptive pretraining (TAPT) (Gururangan et al., 2020) and performs transfer learning after selecting the best source languages based on their cross-lingual transfer score. For a direct comparison, we also apply TAPT and use GradSim to group languages for multilingual training. As shown in Table 2, GradSim sets the new state of the art on AfriSenti. It is superior to the previous approach of Wang et al. (2023) that only considers the pairwise transfer scores, neglecting possible interactions of different source languages. Instead, GradSim maximizes the overall gradient similarities from a global perspective.

**Sequence tagging.** Table 3 provides our results for multilingual named entity recognition. We report the state-of-the-art results from Shaffer (2021) as baseline results. Our approach GradSim outperforms the prior state of the art on most high-resource languages and all low-resource languages, again leading to the best overall results.

Our results for POS tagging are provided in Table 4. GradSim outperforms multilingual and monolingual training without language grouping as well as language grouping based on other metrics. It performs best on average over the low-resource languages as well as on average over all languages.

Given the results on sequence tagging tasks, we find that low-resource languages benefit more from language grouping. For high-resource languages, additional training sources from other languages

| NER | Multi. | Mono. | Family | Embed. (prior) | GradSim (ours) |
|---|---|---|---|---|---|
| *high-resource* | | | | | |
| ar* | 86.65 | 85.25 | 84.92 | 85.25 | **88.02** |
| he | 84.21 | 84.51 | 82.47 | **84.83** | 84.06 |
| da* | 90.00 | 87.57 | 89.64 | 90.49 | **91.65** |
| de | 84.42 | 82.42 | 84.18 | 85.73 | **87.27** |
| en | 81.97 | 77.91 | 81.28 | **83.37** | 83.31 |
| es* | 89.59 | 82.01 | 88.87 | 89.90 | **90.85** |
| fr | 88.22 | 82.83 | 87.78 | **89.79** | 89.54 |
| hi* | 87.30 | 84.04 | 85.51 | 87.17 | **88.25** |
| it | **92.27** | 86.03 | 88.60 | 90.52 | 90.72 |
| ru* | 88.32 | 88.18 | 87.77 | 88.55 | **88.70** |
| ko | 85.97 | 86.54 | 84.66 | **86.91** | 85.92 |
| ja* | 71.08 | 66.83 | 66.83 | 71.40 | **75.56** |
| zh | **79.36** | 73.66 | 73.66 | 79.12 | 75.49 |
| *avg** | 85.34 | 82.14 | 83.55 | 85.62 | **86.10** |
| *low-resource* | | | | | |
| sw* | 88.22 | 63.30 | 55.11 | 90.13 | **90.22** |
| yo* | 77.24 | 7.74 | 21.81 | 85.33 | **86.22** |
| *avg** | 82.73 | 35.52 | 38.46 | 87.73 | **88.22** |
| **avg (all)*** | 84.99 | 75.92 | 77.54 | 85.90 | **86.39** |

Table 3: Results on WikiAnn, a NER benchmark, in micro F1. The numbers of the four baseline / previous state-of-the-art methods are taken from Shaffer (2021) and micro-averaged over the different classes. * indicates the settings with statistically significant improvements using GradSim compared to Embed. (prior), the second-best system. Further baseline results are given in Table 10 in Appendix B for space reasons.

| | Multi. | Mono. | Family | Embed. (FT) | GradSim (ours) |
|---|---|---|---|---|---|
| *high-resource* | | | | | |
| bg* | **99.42** | 99.34 | 99.21 | 99.38 | 99.40 |
| cs* | 99.00 | 99.00 | 98.91 | 98.99 | **99.01** |
| da* | 98.22 | **98.66** | 98.12 | 98.06 | 98.55 |
| de | 94.47 | **94.72** | 94.43 | 94.59 | 94.43 |
| en | 97.06 | **97.34** | 96.88 | 97.25 | 97.18 |
| es | **97.35** | 97.29 | 97.18 | 97.23 | 97.21 |
| eu | 95.95 | 96.01 | 96.01 | 96.04 | **96.09** |
| fa* | 97.30 | 97.31 | 97.20 | 97.35 | **97.41** |
| fi* | 97.51 | 97.64 | 97.61 | 97.51 | **97.70** |
| fr* | 96.58 | 96.48 | 96.21 | 96.40 | **96.67** |
| he* | 97.39 | 97.25 | 97.25 | 97.31 | **97.43** |
| hi | 97.56 | 97.62 | 97.49 | **97.64** | 97.55 |
| hr | 97.66 | **97.67** | 97.56 | 97.65 | 97.57 |
| id | 91.16 | **91.78** | 91.78 | 91.56 | 91.20 |
| it* | 98.60 | **98.68** | 98.45 | 98.51 | 98.58 |
| nl* | 93.76 | **93.94** | 93.67 | 93.80 | 93.88 |
| no* | 98.88 | **99.03** | 98.91 | 98.95 | 99.01 |
| pl* | **98.58** | 98.56 | 98.42 | 98.47 | 98.52 |
| pt* | 98.49 | 98.51 | 98.43 | 98.44 | **98.54** |
| sl* | 98.94 | **99.03** | 98.90 | 98.96 | 99.02 |
| sv | **98.86** | 98.71 | 98.77 | 98.81 | 98.81 |
| *avg** | 97.27 | **97.36** | 97.21 | 97.28 | 97.32 |
| *low-resource* | | | | | |
| el* | 98.56 | 98.41 | 98.20 | 98.57 | **98.59** |
| et* | 95.34 | 94.76 | 95.46 | 95.27 | **95.78** |
| ga* | 93.34 | 92.49 | 93.32 | 93.10 | **93.52** |
| hu | 96.82 | 96.87 | 97.01 | **97.14** | 96.94 |
| ro | **95.74** | 94.65 | 95.32 | 95.74 | 95.14 |
| ta* | 85.63 | 84.55 | 84.55 | 87.16 | **88.32** |
| *avg** | 94.24 | 93.62 | 93.98 | 94.50 | **94.72** |
| **avg (all)*** | 96.60 | 96.53 | 96.49 | 96.66 | **96.74** |

Table 4: Results on the UD POS tagging dataset in accuracy. * indicates the settings with statistically significant improvements using GradSim compared to Embed. (FT), the second-best system. Further baseline results are provided in Table 11 in Appendix B.

have a less prominent impact when enough in-language training data is available. It highlights the value of multilingual learning with well-suited languages to enhance the performance of low-resource languages, providing a key strategy for advancing future low-resource NLP research.

**Significance tests.** We run permutation-based significance tests following Dror et al. (2018) with a significance level of 0.05 between GradSim and the respective second-best system on all three datasets. In Tables 1, 3 and 4, settings with statistically significant improvements when using GradSim are marked with *. The results show that GradSim is significantly better than the second-best system in 32 out of 37 single language settings where GradSim outperforms the second-best system across three datasets. In addition, its average performance across all languages is significantly better than the other systems on all three datasets.

# 5 Analysis

To analyze the behavior of the model, we perform the following analyses on the AfriSenti dataset: A qualitative analysis of the data in order to better un-

derstand differences coming from data peculiarities (Section 5.1), a correlation analysis to explain why some grouping methods work better than others (Section 5.2), and an ablation study to investigate the impact of our design choices (Section 5.3).

## 5.1 Topic Analysis

Although data analysis is valuable for research progress, it is challenging for foreign languages. Therefore, we choose a semi-automatic approach involving machine translation and manual inspection for better understanding the input data of our models: For each language, we first extract the 50 most relevant keywords via a term frequency-inverse document frequency (TF-IDF) method. Then, we use the *pygoogletranslate* API[5] to translate the keywords into English and remove duplicate words and stop words. Table 6 provides exem-

---
[5] https://github.com/Saravananslb/py-googletranslation

| Grouping methods | Pearson correlation coefficient | | |
| --- | --- | --- | --- |
| | ↔ transfer score | ↔ typological similarity | ↔ topic similarity |
| *Baseline* | | | |
| Cross-Transfer Score (oracle) | 1 | 0.353 | 0.5079 |
| Language Family | 0.0869 | 0.5278 | 0.2680 |
| Typological similarity | 0.3530 | 1 | 0.0353 |
| Embedding distance (PLM) | 0.4029 | 0.7696 | -0.2383 |
| Embedding distance (FT) | 0.4667 | 0.7240 | -0.0252 |
| *Gradient similarity wrt. different layers (from deep to shallow)* | | | |
| Classification layer | **0.6963** | 0.3944 | **0.4749** |
| Encoder layer 23 | 0.6485 | 0.6377 | 0.1486 |
| Encoder layer 21 | 0.5526 | 0.7811 | 0.1134 |
| Encoder layer 18 | 0.4462 | 0.8181 | -0.0601 |
| Encoder layer 15 | 0.4602 | 0.8329 | 0.1083 |
| Encoder layer 12 | 0.4532 | 0.8566 | -0.0731 |
| Encoder layer 6 | 0.4542 | 0.8586 | -0.0342 |
| Encoder layer 0 | 0.4526 | 0.8526 | 0.0721 |

Table 5: Results of our correlation analysis on the AfriSenti dataset.

| Language (family) | Keywords |
| --- | --- |
| Swahili (Niger-Congo) | god, thank you, major, national, minister, better, package, service, continue, dr, education, citizens, news, world, construction, people, region, police, state, president, father, army |
| Amharic (Afro-Asiatic) | flower, city, season, a matter, government, discussion, december, press release, district, information, administration, public, government, the racist, man, poison |
| Xitsonga (Niger-Congo) | mozambique, listen, wake up, awake, live, conform, sugar, home, lake, leave, speed, connect, come |

Table 6: Exemplary keywords from AfriSenti tweets of different languages (reduced to a subset for space reasons, full set is provided in Appendix D (Table 15)).

plary results for three languages of the AfriSenti dataset. The complete set of keywords for all languages is provided in Appendix D (see Table 15).

While the keywords extracted for Swahili (*sw*) and Amharic (*am*) are mainly centered around political and administrative topics, e.g., national, minister, education, government etc, the keywords for Xitsonge (*ts*) are more related to every-day life aspects. The multilingual model performance reveals that indeed Swahili and Amharic can effectively be trained together while Swahili and Xitsonga rather harm each other, even though Swahili and Xitsonga belong to the same language family and Swahili and Amharic do not. When looking at the language grouping results, GradSim indeed groups *sw* and *am* together (see Table 12 in Appendix D), thus, is able to capture their topical similarity, while a

language family-based grouping would cluster *sw* and *ts* into the same group.

## 5.2 Correlation Analysis

Table 5 provides the results of our correlation analysis that we perform on the AfriSenti dataset. In particular, we compute the Pearson correlation coefficient between the different grouping methods (similarity measures) that we study and different characteristics, such as model-specific characteristics (measured by cross-lingual transfer score), language-specific characteristics (measured by typological similarity) and topic-specific characteristics (measured by keyword embedding distance).

From the results, we can draw a number of interesting conclusions, namely:

(i) The transfer score is not correlated with language family information and only weakly correlated with embedding-based similarity measures often used in related work. For gradient similarity, we see considerably higher correlation values, supporting our proposal of using this similarity measure for language grouping.

(ii) There is a relatively weak correlation between the cross-lingual transfer score and the typological similarity, while language family and embedding-based similarity measures show a high correlation with typological language similariy. This indicates that these similarity measures capture the linguistics-based language information well, which, however, does not translate into better transfer performance. Similar to the oracle measure (transfer score), gradient similarity based on the classifier parameters is only weakly correlated

with typological language similarity.

(iii) Based on the keywords extracted for our analysis in Section 5.1, we retrieve keyword embeddings from the pretrained model encoder and average them for each language. We then compare the similarities of the keyword-based language embeddings with our different similarity measures using Pearson correlation. We find that they are only weakly correlated with language family information and even weakly negatively correlated with embedding distances. However, the correlation with the cross-transfer score and our proposed gradient similarity is larger, indicating that the gradient similarity can indeed pick up the topic information of the data.

(iv) While higher layers are higher correlated with task performance, lower layers show a higher correlation with typological distance. This indicates that lower layers encode rather general language-specific information while higher layers capture task-related information.

### 5.3 Ablation Study

**Gradients from different layers.** Table 7 shows an ablation study of our model. The main design choice of our approach is the position in the model where to take the gradients. In our analysis, we compare model performance when using gradients from different layers. We see a clear trend that higher layers are better suited than lower layers. In particular, taking the gradients directly from the classification layers leads to the best results.

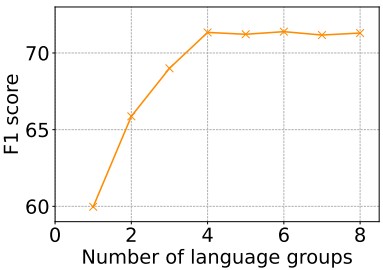

Figure 3: Ablation study of different number of groups on AfriSenti: average F1 w.r.t. number of groups.

**Number of language groups.** For our experiments, we choose the number of language groups $K$ to be the same as the number of language families covered in the datasets.[6] However, $K$ is a hyperparameter of our method. Therefore, we in-

---

[6]Except for WikiAnn, where we set $K$ to the same number as prior work (Shaffer, 2021) for a fair comparison.

vestigate the performance for $K \in \{1 \ldots 8\}$ in Figure 3. Choosing $K = 2$ can already improve the performance compared to purely multilingual training ($K = 1$). Until $K = 4$, the performance further improves and then converges for larger $K$.

| Gradient from layer | Task performance |
|---|---|
| Classification layer | **71.34** |
| Encoder layer 23 | 69.95 |
| Encoder layer 21 | 69.43 |
| Encoder layer 18 | 69.21 |
| Encoder layer 15 | 68.91 |
| Encoder layer 12 | 69.19 |
| Encoder layer 6 | 69.19 |
| Encoder layer 0 | 68.22 |

Table 7: Ablation study of gradients from different layers on the AfriSenti dataset.

## 6 Discussion

In this section, we summarize our main findings.

**Language similarity is not enough to determine transfer suitability.** When sharing knowledge across languages, the information about linguistics-based language similarity (e.g., whether the languages come from the same language family or how languages are typologically similar to each other) is not enough for optimal performance. This observation is in line with the findings by Tan et al. (2019), Shaffer (2021) and Malkin et al. (2022) that languages from the same family may still exhibit distinct linguistic features and, thus, language-family based grouping can enhance model performance only to a certain extent. In addition, we find that there are other aspects that will affect multilingual model performance and, therefore, need to be taken into account, such as the topical distribution of the data.

**Gradient-based method does not require any prior knowledge.** Our proposed gradient-based approach for grouping languages is a pure model-based approach, thus, does not require any prior knowledge about the language, task or data. As a result, it can be successfully applied, even when the data distribution (e.g., topical distribution) is unknown (e.g., because we are dealing with foreign languages). While our current work only presents results for language grouping for multilingual models, the method itself is more general and can be applied to other settings as well, such as multi-task learning or multi-domain setups.

**Lower layers capture language, upper layers task information.** Adding to previous work on analyzing transformer-based pretrained language models (Raganato and Tiedemann, 2018; Jawahar et al., 2019; Tenney et al., 2019), our correlation analysis shows that gradient similarity between languages from lower layers are more correlated to language-specific distances, i.e., low layers seem to encode language-specific information, while gradient similarity from upper layers are more correlated to task-specific performance, i.e., upper layers tend to capture task-specific information.

## 7 Conclusion

In this paper, we addressed the challenging problem of grouping languages for effective multilingual training. We proposed a gradient-based grouping approach and showed in our experiments that it is better correlated to cross-lingual transfer performance than language family or language embedding-based grouping. In our analysis, we identified topical distribution differences as one potential challenge that can be addressed effectively by our approach. Furthermore, our correlation analysis confirmed results from prior work that lower layers of transformer-based pretrained models seem to encode language-specific features, while upper layers capture task-specific information. Our method shows superior performance compared to a variety of baseline methods for language grouping on three diverse datasets and, in particular, sets the new state of the art on a multilingual sentiment analysis benchmark dataset consisting of low-resource African languages.

## Limitations

One limitation of our work is the scope of evaluation. While we performed experiments on three diverse text classification and sequence tagging tasks, GradSim is generally applicable to a wide range of tasks and could thus be evaluated on even further tasks.

Besides, our experiments currently focus on multilingual settings and datasets. Experiments for multi-domain and multi-task settings are outside the scope of this work, however, an interesting direction for future work.

Finally, compared to the large number of languages in the world, the set of languages in our work is still limited and, thus, our results might not be representative for all languages of the world.

However, we chose the datasets for our experiments with the aim of covering a broad variety of languages, including African languages which are typically under-explored in NLP research.

## Ethics Statement

Our work focuses on multilingual and low-resource settings. For instance, we investigate our models on African languages which are typically under-represented and under-explored in NLP research. Including them into NLP research is important from an ethical point of view.

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

## A  Dataset Information

Table 8 provides statistics of AfriSenti, the benchmark dataset we chose for our text classification experiments. Table 9 gives language information for the WikiAnn and Universal Dependencies datasets. We selected 15 languages from WikiAnn for NER following prior work (Shaffer, 2021) and 27 languages for POS tagging following Yasunaga et al. (2018); Lange et al. (2021a).

| Family | Language | Dataset | | |
|---|---|---|---|---|
| | | Train | Dev | Test |
| Afro-Asiatic | Amharic (am) | 5,985 | 1,498 | 2,000 |
| | Algerian Arabic (dz) | 1,652 | 415 | 959 |
| | Hausa (ha) | 14,173 | 2,678 | 5,304 |
| | Moroccan Arabic (ma) | 5,584 | 1,216 | 2,962 |
| English-Creole | Nigerian Pidgin (pcm) | 5,122 | 1,282 | 4,155 |
| Indo-European | Mozambican Portuguese (pt) | 3,064 | 768 | 3,663 |
| Niger-Congo | Igbo (ig) | 10,193 | 1,842 | 3,683 |
| | Kinyarwanda (kr) | 3,303 | 828 | 1,027 |
| | Swahili (sw) | 1,198 | 454 | 749 |
| | Xitsonga (ts) | 805 | 204 | 255 |
| | Twi (twi) | 3,482 | 389 | 950 |
| | Yoruba (yo) | 8,523 | 2,091 | 4,516 |

Table 8: Language information and dataset statistics of AfriSenti. 12 African languages from 4 language families are included in our study.

## B  Additional Baseline Experimental Results

Here we provide the experimental results on the WikiAnn and UD POS tagging dataset with grouping methods based on cross-lingual transfer score, typological language similarity and language embeddings distance.

Comparing GradSim to other baseline methods in Table 10 and 11 , we can draw similar conclusions as in Section 4.4: First, GradSim achieves the best average performance and performs especially well in low-resource languages, revealing the importance of multilingual learning with suitable sets of source languages for low-resource languages. Additionally, GradSim comes very close to the oracle upper bound without any prior knowledge about the cross-lingual transfer performance.

## C  Language Grouping Reults

In this section, we provide further details in Table 12, 13 and 14 of our language grouping results with the best language groups based on different similarity measures on the three datasets.

**WikiAnn NER dataset**

| Family | Language |
|---|---|
| Afro-Asiatic | Arabic (ar) |
| | Hebrew (he) |
| Indo-European | Danish (da) |
| | German (de) |
| | English (en) |
| | Spanish (es) |
| | French (fr) |
| | Hindi (hi) |
| | Italian (it) |
| | Russian (ru) |
| Niger-Congo | Swahili (sw) |
| | Yoruba (yo) |
| Koreanic | Korean (ko) |
| Japonic | Japanese (ja) |
| Sino-Tibetan | Mandarin (zh) |

**UD POS tagging**

| Family | Languages |
|---|---|
| Indo-European | Bulgarian (bg) |
| | Czech (cd) |
| | Danish (da) |
| | German (de) |
| | Greek (el) |
| | English (en) |
| | Spanish (es) |
| | Persian (fa) |
| | French (fr) |
| | Irish (ga) |
| | Hindi (hi) |
| | Croatian (hr) |
| | Italian (it) |
| | Dutch (nl) |
| | Norwegian (no) |
| | Polish (pl) |
| | Portuguese (pt) |
| | Romanian (ro) |
| | Slovenian (sl) |
| | Swedish (sv) |
| Basque | Basque (eu) |
| Uralic | Finnish (fi) |
| | Estonian (et) |
| | Hungarian (hu) |
| Afro-Asiatic | Hebrew (he) |
| Austronesian | Idonesian (id) |
| Dravidian | Tamil (ta) |

Table 9: Language information of WikiAnn and Universal Dependencies datsets for NER and POS tagging tasks, respectively.

## D  Keyword extraction results

In Table 15, we give the full set of keywords extracted from the AfriSenti dataset of 12 African languages.

| | Oracle upper bound | Typological similarity | Embedding dis (PLM & FT) | GradSim (ours) |
|---|---|---|---|---|
| *high-resource* | | | | |
| ar | 88.43 | 87.40 | **88.38** | 88.02 |
| he | 85.22 | 84.12 | **84.88** | 84.06 |
| da | 92.36 | **92.22** | 92.17 | 91.65 |
| de | 88.24 | **88.01** | 87.89 | 87.27 |
| en | 83.98 | **83.75** | 83.42 | 83.31 |
| es | 91.58 | **91.92** | 91.38 | 90.85 |
| fr | 89.72 | **90.13** | 89.83 | 89.54 |
| hi | 88.99 | 86.20 | **89.00** | 88.25 |
| it | 91.13 | **91.16** | 90.84 | 90.72 |
| ru | 88.70 | **88.71** | 88.39 | 88.70 |
| ko | 87.03 | 84.61 | **86.99** | 85.92 |
| ja | 67.74 | 67.80 | 67.33 | **75.56** |
| zh | 76.43 | **76.72** | 75.93 | 75.49 |
| *avg* | 86.12 | 85.60 | 85.88 | **86.10** |
| *low-resource* | | | | |
| sw | 90.37 | 78.79 | 89.70 | **90.22** |
| yo | 86.22 | 61.91 | 6.20 | **86.22** |
| *avg* | 88.30 | 70.35 | 47.95 | **88.22** |
| **avg (all)** | 86.41 | 83.56 | 80.82 | **86.39** |

Table 10: Additional experimental results on the WikiAnn dataset. We compare the grouping performance based on different language simialrity measures. We have the same language grouping results for Embedding distance (PLM) and Embedding distance (FT) (See Table 13) and merge them in one column.

| | Oracle upper bound | Typological similarity | Embedding distance (PLM) | Embedding distance (FT) | GradSim (ours) |
|---|---|---|---|---|---|
| *high-resource* | | | | | |
| bg | 99.42 | **99.40** | 99.35 | 99.37 | **99.40** |
| cs | 99.01 | 98.99 | **99.01** | 98.99 | **99.01** |
| da | 98.56 | 98.04 | 98.09 | 98.07 | **98.55** |
| de | 94.68 | **94.61** | 94.36 | 94.55 | 94.43 |
| en | 97.36 | 96.92 | 97.16 | **97.26** | 97.18 |
| es | 97.36 | **97.35** | 97.23 | 97.25 | 97.21 |
| eu | 96.06 | 96.01 | 95.98 | 96.07 | **96.09** |
| fa | 97.34 | 97.24 | 97.35 | 97.33 | **97.41** |
| fi | 97.66 | 97.50 | 97.63 | 97.53 | **97.70** |
| fr | 96.59 | 96.40 | 96.51 | 96.45 | **96.67** |
| he | 97.44 | 97.24 | 97.29 | 97.30 | **97.43** |
| hi | 97.60 | 97.60 | 97.61 | **97.63** | 97.55 |
| hr | 97.68 | 97.40 | **97.74** | 97.65 | 97.57 |
| id | 91.36 | **91.75** | 91.36 | 91.59 | 91.20 |
| it | 98.60 | 98.56 | **98.58** | 98.51 | **98.58** |
| nl | 94.01 | 93.53 | 93.72 | 93.79 | **93.88** |
| no | 99.04 | 98.91 | 98.92 | 98.96 | **99.01** |
| pl | 98.59 | **98.54** | 98.49 | 98.49 | 98.52 |
| pt | 98.56 | **98.56** | 98.49 | 98.42 | 98.54 |
| sl | 99.03 | 98.87 | **99.02** | 98.96 | **99.02** |
| sv | 98.74 | **98.88** | 98.80 | 98.83 | 98.81 |
| *avg* | 97.37 | 97.25 | 97.27 | 97.29 | **97.32** |
| *low-resource* | | | | | |
| el | 98.62 | 98.42 | 98.50 | 98.56 | **98.59** |
| et | 95.58 | 95.65 | 95.04 | 95.26 | **95.78** |
| ga | 93.50 | 93.46 | 93.28 | 93.08 | **93.52** |
| hu | 96.99 | 96.90 | **97.39** | 97.14 | 96.94 |
| ro | 95.20 | **95.92** | 94.76 | 95.79 | 95.14 |
| ta | 88.33 | 86.16 | 88.04 | 87.02 | **88.32** |
| *avg* | 94.70 | 94.42 | 94.50 | 94.48 | **94.72** |
| **avg (all)** | 96.77 | 96.62 | 96.66 | 96.66 | **96.74** |

Table 11: Additional experimental results on the UD POS tagging dataset.

| Method | | Language groups |
|---|---|---|
| Oracle upper bound | group 0 | **am, ha, kr, ma, pt, sw** |
| | group 1 | **dz, pcm** |
| | group 2 | **ig** |
| | group 3 | pcm, **ts, twi, yo** |
| Language Family | group 0 | **am, dz, ha, ma** |
| | group 1 | **ig, kr, sw, ts, twi, yo** |
| | group 2 | **pcm** |
| | group 3 | **pt** |
| Typological dis. | group 0 | **am, dz, ha, ig, ma, sw, twi, yo** |
| | group 1 | **kr** |
| | group 2 | **pcm, ts** |
| | group 3 | **pt** |
| Embedding dis. (PLM) | group 0 | **am, dz, ma, pt,** twi |
| | group 1 | **ha, ig, kr, pcm, ts, twi** |
| | group 2 | **sw** |
| | group 3 | **yo** |
| Embedding dis. (FT) | group 0 | **am, kr, pt, sw** |
| | group 1 | **dz, ma, pcm** |
| | group 2 | **ha, ig, ma, ts, yo** |
| | group 3 | **ma, ts, twi** |
| GradSim (ours) | group 0 | **am, ha, kr, ma, pt, sw** |
| | group 1 | **dz, pcm, ts, twi** |
| | group 2 | **ig** |
| | group 3 | **yo** |

Table 12: Language grouping results on AfriSenti dataset. Unbolded languages are the source-only languages in the group, i.e., they only participate in the multilingual training but are evaluated in another language group during inference (details see Section 3.2).

| Method | | Language groups |
|---|---|---|
| Oracle upper bound | group 0 | **ar, fr, sw, yo** |
| | group 1 | **he, da, de, en, es, hi, it, ko** |
| | group 2 | **ja** |
| | group 3 | **ru, zh** |
| Language Family | group 0 | **ar, he** |
| | group 1 | **da, de, en, es, fr, hi, it, ru** |
| | group 2 | **sw, yo** |
| | group 3 | **ko** |
| | group 4 | **ja** |
| | group 5 | **zh** |
| Typological dis. | group 0 | **ar, he** |
| | group 1 | **da, de, en, es, fr, it, ru** |
| | group 2 | **sw, yo** |
| | group 3 | **hi, ko, ja, zh** |
| Embedding dis. (PLM) | group 0 | **ar, en, es, fr, hi, it, sw** |
| | group 1 | **he, da, de, ru, ko** |
| | group 2 | **ja, zh** |
| | group 3 | **yo** |
| Embedding dis. (FT) | group 0 | **ar, en, es, fr, hi, it, sw** |
| | group 1 | **he, da, de, ru, ko** |
| | group 2 | **ja, zh** |
| | group 3 | **yo** |
| GradSim (ours) | group 0 | **ar, en, es, sw, yo** |
| | group 1 | **he, da, de, ko, ja** |
| | group 2 | **fr, hi, it, ru** |
| | group 3 | **zh** |

Table 13: Language grouping results on the WikiAnn dataset. The 15 languages we study covers 6 language families, while we use $K = 4$ numbers of groups for our experiment following prior work Shaffer (2021) for a fair comparison.

| Method | | Language groups |
|---|---|---|
| Oracle upper bound | group 0 | **bg, de, fa, pl, sl, sv, el** |
| | group 1 | **cs, da, en, es, fr, hr, it, no, pt, ga, ro, eu, fi, et, hu, he** |
| | group 2 | **hi** |
| | group 3 | **id** |
| | group 4 | **nl** |
| | group 5 | **ta** |
| Language Family | group 0 | **bg, cs, da, de, en, es, fa, fr, hi, hr, it, nl, no, pl, pt, sl, sv, el, ga, ro** |
| | group 1 | **eu** |
| | group 2 | **fi, et, hu** |
| | group 3 | **he** |
| | group 4 | **id** |
| | group 5 | **ta** |
| Typological dis. | group 0 | **bg, cs, da, de, en, es, fi, fr, hr, it, nl, no, pl, pt, sl, sv,el, et, ga, hu, ro** |
| | group 1 | **eu** |
| | group 2 | **fa** |
| | group 3 | **he** |
| | group 4 | **id** |
| | group 5 | **hi, ta** |
| Embedding dis. (PLM) | group 0 | **bg,nl** |
| | group 1 | **cs, da, de, en, it, no, pl, pt, sv, fi, id** |
| | group 2 | **es, fa, hi, hr, el** |
| | group 3 | **fr, he, ta** |
| | group 4 | **ga, eu, et** |
| | group 5 | **sl, ro, hu** |
| Embedding dis. (FT) | group 0 | **bg, de, fa, nl, pl, sl, sv, el** |
| | group 1 | **cs, da, hr, it, no, ro, hu** |
| | group 2 | **en, es, fr, pt, he** |
| | group 3 | **hi** |
| | group 4 | **ga, eu, fi, et** |
| | group 5 | **id, ta** |
| GradSim (ours) | group 0 | **bg, cs, pt, ro, eu, hu, he** |
| | group 1 | **da, de, sv, fi, et** |
| | group 2 | **en, es, fr, hi, it, id** |
| | group 3 | **fa, pl, el** |
| | group 4 | **hr, nl, ta** |
| | group 5 | **no, sl, ga** |

Table 14: Language grouping results on the Universal Dependency POS tagging dataset.

| Languages | Keywords |
|---|---|
| am | flower, city, season, a matter, government, discussion, december, press release, district, information, administration, public, government, the racist, man, poison |
| dz | advantage, god, on my mind, welcome, my lord, dramatically, from, i swear, the people, , sugar, complain, we manage, overwhelming, i was overwhelmed, wicked, the incident, need, cash, poor |
| ha | ameen, safe, bring, amen, lord, bless you, fight, money, now, ai, whether, month, nonsense |
| ig | love, good, bless, husband, god, happy, birthday, leader, glory, king, thank you, sustain, glory, jesus, money, people, crazy, power, mouth, people, problem, dug, pieces, down, stupid, shut up, devil, fear, kill, call, poison, anger |
| kr | very, good, Rwanda, god, good, peace, day, best, thank you, lord, together, comfort, under, discussion, news, plan, president, problem, person', bad, child, now, woman |
| ma | god, justice, for you, thanks, upon you, amazing, regards, good luck, congrats, blessed, development, one thousand, cave, between, facing, good, awake, president, facing, good, awake, president, special, causes, minister, film, desert, causes, minister, film, desert |
| pcm | you, love, fine, god, sweet, my, baby, fit, bless, thank, today, hustle, rush, happy, enjoy, like, even, life, no, say, like, person, people, pain, go, even |
| pt | no, say, like, them, again, person, want, people, you all, use, pain, why, god, lord, eternal, gospel, sin, Christ, church, earth, repentance, Jesus, name, message, June, worse, nothing, people, unfortunately, problem |
| sw | god, thank you, major, national, minister, better, package, service, continue, dr, education, citizens, news, world, construction, people, region, police, state, president, father, army |
| ts | mozambique, listen, wake up, awake, live, conform, sugar, home, lake, leave, speed, connect, come |
| twi | mother, god, thousand, money kill, sleep, yes, a little, even, why, who, stop, team, good, father, word, say, that |
| yo | all, god, give, good, day, lord, come, amen, how, that, know, put, son, word, may, want, who, her, become, go |

Table 15: Keyword extraction results.