# OpenReview forum: "GradSim: Gradient-Based Language Grouping for Effective Multilingual Training"
_EMNLP/2023/Conference — EMNLP 2023 Main_

### Official Review · Reviewer_5L1B · 2023-07-31

**Soundness:** 3

**Excitement:**

3: Ambivalent: It has merits (e.g., it reports state-of-the-art results, the idea is nice), but there are key weaknesses (e.g., it describes incremental work), and it can significantly benefit from another round of revision. However, I won't object to accepting it if my co-reviewers champion it.

**Missing References:**

L046
A few previous works analyzed the impact of typological features on cross-lingual transfer, e.g., Pires et al. 2019

L515
Malkin et al. 2022, explore other aspects affecting cross-lingual transfer: sharing writing-system and pre-training data size.

**Paper Topic And Main Contributions:**

The past results have shown that in the face of scarcity of in-language data for downstream tasks, it’s beneficial to fine-tune the model with the data in multiple languages.

This work introduces a method for selecting language groups for multilingual training. The first step of the approach is multilingual training on all available data. The main step is computing average gradients from the backpropagation of training examples in specific languages. Then similarity between languages is defined as cosine between their averaged gradients. Lastly, a predefined number of language groups is obtained by solving the set cover problem (aimed to minimize similarity between languages in each group).

The authors show that their method of fine-tuning is competitive for low-resource languages in different downstream tasks: AfriSenti, WikiAnn, and UD. The paper addresses the important issue of effectively utilizing cross-lingual transfer for low-resource languages. However, as I read it, the authors overstate the benefits of their solution. Based on the presented results, I suspect that the presented method may not be significantly better than the baselines.

**Questions For The Authors:**

A. Can your method also be applied for pre-training?

B. Did you consider assigning some languages to multiple groups? For instance, when they have gradients similar to many languages in different groups?

C. Regarding the finding iv) in 5.2. The difference between layers in correlations with task performance and typological distance may be the result of fine-tuning. When computing gradients, were the layers of the encoder frozen or updated in this experiment? I would suggest doing the former in an attempt to interpret the information encoded throughout the layers.

D. Why the results for the embedding-based baseline are not shown in Table 4? It looks like there’s enough space to add them.

**Reasons To Accept:**

- Well-motivated research problem about finding the optimum grouping of languages for multilingual training. Especially aimed to benefit low-resource languages.

- I appreciate the analysis exploring the details of knowledge transferability to explain the proposed method's effectiveness.

**Reasons To Reject:**

- The authors show neither standard deviations nor the significance of differences between the analyzed methods. It’s suspicious, especially given that the experiments were run with 5 random seeds (claimed in L287)

- I don’t agree with the author’s claim that the method is data agnostic (claimed in L520). Clearly, training data for downstream tasks are required to compute language-specific gradients.

- The authors state that their method is more efficient than an “Oracle method” based on a cross-lingual transfer map. However, this baseline requires the same number of training steps (in N monolingual training instead of joint multilingual training). Admittedly, the “Oracle method” requires N x (N - 1) evaluation runs. Still, for the analyzed tasks, the evaluation runs are not computationally demanding.

- The introduction of the methodology is opaque to me. Until section 4, it was not clear that the method was applied for fine-tuning instead of pre-training. Using the term “training” is giving a false impression that it’s the latter.

**Reproducibility:**

4: Could mostly reproduce the results, but there may be some variation because of sample variance or minor variations in their interpretation of the protocol or method.

**Reviewer Confidence:**

5: Positive that my evaluation is correct. I read the paper very carefully and I am very familiar with related work.

**Typos Grammar Style And Presentation Improvements:**

L167 “Du to”  -> “Due to”
L319 “data peculiarities” is not the best word choice IMO
L411 “in the appendix” -> “in Appendix”

---

> ### Author Rebuttal · Authors · 2023-08-28
>
> We thank the reviewer for the detailed comments and suggestions. To address your concerns and questions:
>
> - **“The authors show neither standard deviations nor the significance of differences between the analyzed methods”**
>
>   Thanks for pointing this out. To address your concerns, we have run permutation-based significance tests [1] and included the standard deviation results on all three dataset.
>
>   For the significance test, we did the test between GradSim and the second best system: on AfriSenti and UD POS tagging, the second best system is the embedding (finetune)-based model, on WikiAnn is the model of prior work [2].
>
>   Results are shown below, **bold languages** represent languages on which GradSim outperforms the second best system (refer to Table 1, 3, 4 in the paper), and the ***mark** represents the language with significance $p\ value<0.05$. On most languages where GradSim performs better, we found statistically significant differences (32 out of 37 languages), indicating that GradSim is significantly better than the second best.
>
>   | AfriSenti | | | | | | | |
>   |:---:|:---:|:---:|:---:|:---:|:---:|:---:|---|
>   | **Language** | **overall*** | **am*** | **dz** | **ha** | **ig*** | **kr*** | **ma*** |
>   | **p-value** | 0.0001 | 0.0001 | 0.0549 | 0.0884 | 0.0304 | 0.0001 | 0.0001 |
>   | **Language** | pcm | **pt*** | **sw*** | **ts** | twi | **yo*** |  |
>   | **p-value** | 0.1925 | 0.0035 | 0.0005 | 0.2662 | 0.8890 | 0.0291 |  |
>
>   | WikiAnn | | | | | | | | | |
>   |:---:|:---:|:---:|:---:|:---:|:---:|:---:|:---:|:---:|:---:|
>   | **Language** | **overall*** | **overall-high*** | **ar*** | he | **da*** | **de** | en | **es*** | fr |
>   | **p-value** | 0.0294 | 0.0472 | 0.0343 | 0.8359 | 0.0142 | 0.0050 | 0.7589 | 0.0080 | 0.9739 |
>   | **Language** | **hi*** | it | **ru*** | ko | **ja*** | zh | **overall-low*** | **sw*** | **yo*** |
>   | **p-value** | 0.0488 | 0.0501 | 0.0451 | 0.9999 | 0.0316 | 0.6555 | 0.0400 | 0.0451 | 0.0045 |
>
>   | UD POS | | | | | | | | | | |
>   |:---:|:---:|:---:|:---:|:---:|:---:|:---:|:---:|:---:|:---:|---|
>   | **Language** | **overall*** | **overall-high** | **bg*** | **cs*** | **da*** | de | en | es | **eu** | **fa*** |
>   | **p-value** | 0.0084 | 0.1455 | 0.0017 | 0.0428 | 0.0172 | 0.8261 | 0.7643 | 0.6461 | 0.1289 | 0.0305 |
>   | **Language** | **fi*** | **fr*** | **he*** | hi | hr | id | **it*** | **nl*** | **no*** | **pl*** |
>   | **p-value** | 0.0376 | 0.0463 | 0.0451 | 0.6106 | 0.7896 | 0.5611 | 0.0268 | 0.0468 | 0.0218 | 0.0482 |
>   | **Language** | **pt*** | **sl*** | sv | **overall-low*** | **el*** | **et*** | **ga*** | hu | ro | **ta*** |
>   | **p-value** | 0.0469 | 0.0228 | 0.7268 | 0.0202 | 0.0428 | 0.0351 | 0.0478 | 0.8049 | 0.8228 | 0.0401 |
>
>   Additionally, we also calculate the standard deviation of GradSim and other baseline models. We give the results of GradSim and the best baseline model here, so as not to take up too much space. We will revise our paper to include standard deviations and significance tests between the analyzed methods.
>
>   | AfriSenti | | | | | | | |
>   |:---:|:---:|:---:|:---:|:---:|:---:|:---:|:---:|
>   | **Method** | **overall** | **am** | **dz** | **ha** | **ig** | **kr** | **ma** |
>   | **GradSim** | **71.34±0.16** | **66.11±2.50** | **67.90±1.23** | **79.97±0.68** | **79.55±0.57** | **72.12±0.94** | **53.68±1.78** |
>   | **Embedding(FT)** | 69.62±0.81 | 59.09±5.45 | 67.36±1.28 | 79.78±0.37 | 78.71±0.44 | 69.79±1.57 | 50.01±1.55 |
>   | **Method** | **pcm** | **pt** | **sw** | **ts** | **twi** | **yo** |  |
>   | **GradSim** | 68.40±0.5 | **72.30±0.36** | **63.05±1.09** | **50.36±1.82** | 63.46±2.86 | **75.09±0.51** |  |
>   | **Embedding(FT)** | **68.46±0.22** | 66.43±2.77 | 62.62±0.40 | 49.20±1.80 | **64.01±2.28** | 75.07±0.16 |  |
>
>   | WikiAnn | | | | | | | | |
>   |:---:|:---:|:---:|:---:|:---:|:---:|:---:|:---:|:---:|
>   | **Method** | **overall** | **ar** | **he** | **da** | **de** | **en** | **es** | **fr** |
>   | **GradSim** | **86.39±0.28** | **88.02±0.29** | 84.06±0.19 | **91.65±0.18** | **87.27±0.15** | 83.31±0.23 | **90.85±0.09** | 89.54±0.12 |
>   | **Prior Work** | 85.90±(-) | 85.25±0.23 | **84.83±0.16** | 90.49±0.09 | 85.73±0.21 | **83.37±0.06** | 89.90±0.21 | **89.79±0.14** |
>   | **Language** | **hi** | **it** | **ru** | **ko** | **ja** | **zh** | **sw** | **yo** |
>   | **GradSim** | **88.25±0.32** | **90.72±0.11** | **88.70±0.18** | 85.92±0.40 | **75.56±4.12** | 75.49±0.08 | **90.22±0.20** | **86.22±0.33** |
>   | **Prior Work** | 87.17±0.42 | 90.52±0.16 | 88.55±0.19 | **86.91±0.18** | 71.40±0.38 | **79.12±0.24** | 90.13±0.43 | 85.33±0.79 |
>
>   (Results of prior work are taken from [2] where the overall standard deviation is not given.)
>
>   | UD POS tagging | | | | | | | |
>   |:---:|:---:|:---:|:---:|:---:|:---:|:---:|:---:|
>   | **Method** | **overall** | **bg** | **cs** | **da** | **de** | **en** | **es** |
>   | **GradSim** | **96.74±0.06** | **99.40±0.03** | **99.01±0.01** | **98.55±0.14** | 94.43±0.07 | 97.18±0.09 | 97.21±0.11 |
>   | **Embedding(FT)** | 96.66±0.03 | 99.37±0.04 | 98.99±0.02 | 98.09±0.08 | **94.56±0.08** | **97.26±0.04** | **97.25±0.10** |
>   | **Method** | **eu** | **fa** | **fi** |  **fr** | **he** | **hi** | **hr** |
>   | **GradSim** | **96.09±0.09** | **97.41±0.11** | **97.70±0.07** | **96.67±0.11** | **97.43±0.04** | 97.55±0.05 | 97.57±0.08 |
>   | **Embedding(FT)** | 96.08±0.06 | 97.33±0.07 | 97.57±0.15 | 96.44±0.11| 97.31±0.04 | **97.62±0.06** | **97.67±0.06** |
>   | **Method** | **id** | **it** | **nl** | **no** | **pl** | **pt** |  **sl** |
>   | **GradSim** | 91.20±0.12 | **98.58±0.04** | **93.88±0.20** | **99.01±0.03** | **98.52±0.05** | **98.54±0.07** | **99.02±0.04** |
>   | **Embedding(FT)** | **91.60±0.13** | 98.52±0.04 | 93.78±0.06 | 98.95±0.03 | 98.49±0.04 | 98.44±0.05 | 98.97±0.03 |
>   | **Method** | **sv** | **el** | **et** | **ga** | **hu** | **ro** | **ta** |
>   | **GradSim** | 98.81±0.01 | **98.59±0.06** | **95.78±0.17** | **93.52±0.27** | 96.94±0.07 | 95.14±0.25 | **88.32±0.17** |
>   | **Embedding(FT)** | **98.84±0.04** | 98.56±0.03 | 95.28±0.34 | 93.02±0.19 | **97.17±0.18** | **95.67±0.29** | 86.81±0.76 |
>
>   [1] Dror, Rotem, et al. "The hitchhiker’s guide to testing statistical significance in natural language processing." Proceedings of the 56th annual meeting of the association for computational linguistics (volume 1: Long papers). 2018.
>
>   [2] Shaffer, Kyle. "Language clustering for multilingual named entity recognition." Findings of the Association for Computational Linguistics: EMNLP 2021. 2021.
>
>
> - **“I don’t agree with the author’s claim that the method is data agnostic (claimed in L520). Clearly, training data for downstream tasks are required to compute language-specific gradients.”**
>
>   Please note that **we didn’t claim ‘data agnosticism’ in the paper** (neither in L520 nor anywhere else).  In L520, we state that “Our proposed gradient-based approach for grouping languages is a pure model- based approach, thus, does not require any prior knowledge about the language, task or data.” Our statement is correct that GradSim is a pure model-based approach: **it can be used even if there is no prior knowledge available about the application setting (task, domain, data etc),** in contrast to grouping methods based on linguistics similarity which require prior knowledge about the training data, e.g., the language family information or typological features.
>
>   Our understanding of ‘data agnosticism’ is that: if a method is not limited to some specific types of data, the method is data-agnostic. **In that respect, GradSim is data-agnostic**, as it is not limited to multilingual data, but can also be applied to multi-task or multi-domain data. Extending our methodology to multi-task or multi-domain settings could be an interesting future work direction.
>
> - **“this baseline requires the same number of training steps (in N monolingual training instead of joint multilingual training). Admittedly, the “Oracle method” requires N x (N - 1) evaluation runs. Still, for the analyzed tasks, the evaluation runs are not computationally demanding.”**
>
>   From the number of evaluation runs, it’s a **$1$ VS $N \times N$** comparison between GradSim and the cross-lingual transfer oracle.
>
>   To measure the gradient similarities across languages, GradSim only required **$1$** joint multilingual training rather than N individual monolingual trainings. We measure the gradients wrt. each language along the optimization process (As described in Section 3.1, L177-L188).
>
>   However, the oracle method always requires **$N \times N$** runs (not $N \times (N-1)$) for $N$ languages. Specifically, taking each language as a target language, we select each of all N languages as the source language (also including the target language itself, therefore it is $N \times N$ rather than $N \times (N-1))$ to form a (source, target) language pair and run a source-to-target cross-lingual transfer experiment. We then use cross-lingual transfer results as the similarity measure between languages for language grouping.
>
>    In practice, on the AfriSenti dataset with 12 languages,  the joint multilingual training of **GradSim takes ~1 hour**, while **the oracle method takes ~17 hours** with the same experimental setup (computational resource, batch size, training epoch etc.). The oracle method will become extremely computationally expensive given a larger number of languages, while GradSim is far more computationally efficient than the oracle method.
>
> - **“Until section 4, it was not clear that the method was applied for fine-tuning instead of pre-training. Using the term “training” is giving a false impression that it’s the latter.”**
>
>   Thanks for your comment, we will clarify this in the updated version of our paper to avoid confusion.
>
> - **Q1: “Can your method also be applied for pre-training?”**
>
>   **Yes, our method can be applied for pretraining.** Based on the same steps described in Section 3, we can first calculate gradient similarities across languages with joint multilingual pretraining (Step 1), then determine the best K language groups based on the similarity measure from the last step (Step 2). Finally, we can pretrain one language model on each language group (Step 3) which can be deployed for downstream fine-tuning tasks. This could be an interesting future work direction.
>
> - **Q2: “Did you consider assigning some languages to multiple groups? ”**
>
>   **Yes, our method is able to assign languages to multiple groups** whenever it's beneficial. For detailed description please refer to Section 3.2 (L206) and 3.3 (L224).
>
> - **Q3: “When computing gradients, were the layers of the encoder frozen or updated in this experiment? ”**
>
>   The encoder layers are updated while computing gradients in our experiments, **following the normal LM fine-tuning setup.** We agree that the difference between layers in correlations with task performance and typological distance may be the result of fine-tuning, which is not a problem from our point of view. In the end we want to study how the deployed model for inference (the encoder of which is fine-tuned) encodes the information.
>
>   We ran new experiments with the frozen encoder and found the overall tendency of correlation across layers holds, i.e., lower layers of transformer models encode language-specific features while higher layers capture task-specific information. Below are the results of the correlation analysis:
>
>   |  | Encoder frozen |  |  | Encoder finetuned |  |  |
>   |:---:|:---:|:---:|:---:|:---:|:---:|:---:|
>   | | Transfer score | Typological distance | Topic similarity | Transfer score | Typological distance | Topic similarity |
>   | Classifier | 0.6817 | 0.3483 | 0.4163 | 0.6963 | 0.3944 | 0.4749 |
>   | Encoder layer-23 | 0.7207 | 0.3851 | 0.0683 | 0.6485 | 0.6377 | 0.1486 |
>   | Encoder layer-21 | 0.7183 | 0.4581 | 0.0916 | 0.5526 | 0.7811 | 0.1134 |
>   | Encoder layer-18 | 0.6810 | 0.6623 | 0.1691 | 0.4462 | 0.8181 | -0.0601 |
>   | Encoder layer-15 | 0.4458 | 0.8101 | 0.0723 | 0.4602 | 0.8329 | 0.1083 |
>   | Encoder layer-12 | 0.4507 | 0.8400 | 0.0362 | 0.4532 | 0.8566 | 0.0362 |
>   | Encoder layer-6 | 0.4541 | 0.8358 | -0.1893 | 0.4542 | 0.8586 | -0.0342 |
>   | Encoder layer-0 | 0.4560 | 0.8458 | -0.1741 | 0.4526 | 0.8526 | -0.0721 |
>
> - **Q4: “Why the results for the embedding-based baseline are not shown in Table 4?”**
>
>   We would have to add two more columns in Table 4 (using the embeddings from the pretrained model and fine-tuned model). Due to space reasons, we only show the results in the appendix. We will think of a way to visualize everything in the updated version.
>
> - **Missing references & Typos**
>
>   Thanks for pointing this out. We will add the missing references and correct the typos in the updated paper.
>
> Again, thank you for your detailed reviews, which are valuable to improving the work. We appreciate that you recognized the motivation of our work, the effectiveness of the proposed method and the value of our correlation analysis which are precisely the main contributions of our work.
>
> We hope our explanation addresses your concerns. Please let us know if you have any further comments. We would appreciate it if you updated your review accordingly and can share your thoughts during the reviewer discussion given our responses to your questions .

---

### Official Review · Reviewer_VWWR · 2023-08-03

**Soundness:** 4

**Excitement:**

4: Strong: This paper deepens the understanding of some phenomenon or lowers the barriers to an existing research direction.

**Paper Topic And Main Contributions:**

This paper proposes a simple step in pretraining multilngual embeddings, namely to cluster languages based on their gradient similarity, and train separate multilingual models for each grouping. Evaluating this new pretraining setup on downstream tasks (sentiment analysis on African languages, NER, POS tagging) a) generally (but not always) results in improvements across different languages compared to the baselines (and on AfriSenti, gets very close to the oracle upper bound result), b) in combination with task-adaptive pretraining, gets better than SoTA on AFriSenti, c) also results in improvements on NER and POS tagging (but less so for high-resource languages on the latter). They look at the correlation between the gradients from different encoder layers and other grouping methods, make the argument that topic information in the corpora is important (more so than typological and linguistic information) and show that their method takes advantage of this more effectively.

**Questions For The Authors:**

- Table 5 looks at the AfriSenti dataset only. I understand the space constraints, but to what extent do these correlations hold across the other tasks that you looked at?
- are all of the averaged F-1 scores micro- or macro-averaged across languages?

**Reasons To Accept:**

- analysis is comprehensive and convincing re: a) incorporation of topical information via their method and b) which layers of the model correspond to linguistic features and which ones correspond to task-specific features.
- small but consistent improvements over corresponding baselines on the tasks evaluated

**Reasons To Reject:**

- improvements on most language pairs for the three tasks are relatively small

**Reproducibility:**

4: Could mostly reproduce the results, but there may be some variation because of sample variance or minor variations in their interpretation of the protocol or method.

**Reviewer Confidence:**

4: Quite sure. I tried to check the important points carefully. It's unlikely, though conceivable, that I missed something that should affect my ratings.

**Typos Grammar Style And Presentation Improvements:**

- line 479: I think you are referring to Table 7 in the text?

---

> ### Author Rebuttal · Authors · 2023-08-28
>
> Thank you for your review and valuable feedback! We are encouraged by your recognition of the value and contribution of our work. We provide answers to your comments and questions below.
>
> - **“improvements on most language pairs for the three tasks are relatively small”**
>
>   GradSim aims at finding the best language groups from a global perspective, i.e., it achieves the highest possible overall performance across all languages rather than on each individual language.
>
>   On Afrisenti, we have achieved an overall improvement of ~ 2 F1 points over the second best methods (embedding-based language grouping). For sequence tagging tasks (NER and POS tagging), the improvement is relatively small as the baseline methods already achieve high performance.
>
>   We have run significance tests and proved that the performance improvement from our approach is statistically significant on most languages (32 out of 37) across the three datasets, detailed results are given in the response to Reviewer 5L1B. We will add these more detailed results in the updated version of our paper.
>
>
> - **Q1: “Table 5 looks at the AfriSenti dataset only. I understand the space constraints, but to what extent do these correlations hold across the other tasks that you looked at?"**
>
>   Thanks for bringing this up. We agree that doing this correlation analysis on other datasets as well would give some interesting information. To address your concerns, we will add the correlation analysis of the other two datasets in our subsequent paper version. We will make use of the additional page in the camera-ready version to do so.
>
> - **Q2: “are all of the averaged F-1 scores micro- or macro-averaged across languages?”**
>
>   For AfriSenti, We use the macro-averaged F1 across languages as the averaged performance, to be consistent with the dataset paper [1]. For the other two datasets (WikiAnn and UD POS tagging), we report the micro-averaged F1 following previous work [2, 3] for a fair comparison. We will clarify this in Section 4.1 in the updated paper.
>
>   [1] Muhammad, Shamsuddeen Hassan, et al. "Afrisenti: A twitter sentiment analysis benchmark for african languages." arXiv preprint arXiv:2302.08956 (2023).
>
>   [2] Shaffer, Kyle. "Language clustering for multilingual named entity recognition." Findings of the Association for Computational Linguistics: EMNLP 2021. 2021.
>
>   [3] Yasunaga, Michihiro, Jungo Kasai, and Dragomir Radev. "Robust multilingual part-of-speech tagging via adversarial training." arXiv preprint arXiv:1711.04903 (2017).
>
> - **Typos**
>
>   Thanks for pointing this out, we will correct that in the updated version of our paper.
>
> We are grateful for your comments and suggestions, which are indeed valuable to improving the work. Hope our explanation addresses your concerns. Please let us know if you have any further comments.
>
> We appreciate the positive feedback you provided and would be grateful if you could share your insights and in-depth understanding of our paper's strengths during the reviewer rebuttal phase.

---

### Official Review · Reviewer_J9J7 · 2023-08-06

**Soundness:** 3

**Excitement:**

3: Ambivalent: It has merits (e.g., it reports state-of-the-art results, the idea is nice), but there are key weaknesses (e.g., it describes incremental work), and it can significantly benefit from another round of revision. However, I won't object to accepting it if my co-reviewers champion it.

**Paper Topic And Main Contributions:**

The authors propose GradSim, a language grouping method based on gradient similarity.They find that the topic distribution of the training data heavily influences multilingual training, and that the lower layers of the converter model encode language-specific features while the higher layers capture task-specific information.

**Reasons To Accept:**

They find that the topic distribution of the training data heavily influences multilingual training, and that the lower layers of the converter model encode language-specific features while the higher layers capture task-specific information. Their  method shows superior performance compared to a variety of baseline methods for language grouping on three diverse datasets and, in particular, sets the new state of the art on a multilingual sentiment analysis benchmark dataset consisting of low-resource African languages.

**Reasons To Reject:**

The authors' experiments were conducted only on text categorization and sequence annotation tasks, lacking experimental comparisons with other e.g. reading comprehension, QA tasks.

**Reproducibility:**

4: Could mostly reproduce the results, but there may be some variation because of sample variance or minor variations in their interpretation of the protocol or method.

**Reviewer Confidence:**

3: Pretty sure, but there's a chance I missed something. Although I have a good feel for this area in general, I did not carefully check the paper's details, e.g., the math, experimental design, or novelty.

---

> ### Author Rebuttal · Authors · 2023-08-28
>
> Thank you for your efforts in reviewing our paper and the recognition of the work’s contribution. Regarding your comment:
>
> - **“The authors' experiments were conducted only on text categorization and sequence annotation tasks, lacking experimental comparisons with other e.g. reading comprehension, QA tasks.”**
>
>   For this work, we chose to ensure depth and robustness on the three selected tasks. Conducting comprehensive experiments across multiple NLP tasks would require significantly more resources and time. Despite our experiments' specificity, the methodology we introduced can certainly be adapted or extended to other tasks like reading comprehension or QA. We believe our findings serve as a valuable foundation on which future research on other multilingual tasks can be built.
>
> Thanks for bringing up this point, which is valuable to further improve and extend our work. Please let us know if you have any further comments.

---

### Meta-Review · Area_Chair_TAWn · 2023-09-17

**Recommendation:** 4

**Metareview:**

This work proposes a method to group languages in multilingual tasks, e.g., text classification and sequence labeling, by leveraging the gradient similarity. The languages are clustered according to the metric, and a model is trained for each cluster.

Strengths
* The proposed method is well-motivated with comprehensive analyses on the impact of topical distributions and representation in layers.
* Experiments were carried out on various tasks with consistent gains especially targeting low-resource language settings.

Weaknesses
* The gains are small, and they have not experimented on other tasks, such as natural language understanding.
* Some clarity issues in the current manuscript, which could be addressed in the revised one without changing the conclusion.

---

### Decision · Program_Chairs · 2023-10-07

**Decision:**

Accept-Main

**Comment:**

This work proposes a method to group languages in multilingual tasks, e.g., text classification and sequence labeling, by leveraging the gradient similarity. The languages are clustered according to the metric, and a model is trained for each cluster.

Strengths
* The proposed method is well-motivated with comprehensive analyses on the impact of topical distributions and representation in layers.
* Experiments were carried out on various tasks with consistent gains especially targeting low-resource language settings.

Weaknesses
* The gains are small, and they have not experimented on other tasks, such as natural language understanding.
* Some clarity issues in the current manuscript, which could be addressed in the revised one without changing the conclusion.